# Applicability of Convolutional Neural Network for Estimation of Turbulent Diffusion Distance from Source Point

**Takahiro Ishigami [1,2,\*], Motoki Irikura [2] and Takahiro Tsukahara [1]**

[1] Department of Mechanical Engineering, Tokyo University of Science, Yamazaki 2641, Noda-shi 278-8510, Japan
[2] Chiyoda Corporation, CGH 4-6-2, Minatomirai, Nishi-ku, Yokohama-shi 220-8765, Japan
\* Correspondence: ishigami.takahiro@chiyodacorp.com; Tel.: +81-45-285-4495

**Abstract:** For locating the source of leaking gas in various engineering fields, several issues remain in the immediate estimation of the location of diffusion sources from limited observation data, because of the nonlinearity of turbulence. This study investigated the practical applicability of diffusion source-location prediction using a convolutional neural network (CNN) from leaking gas instantaneous distribution images captured by infrared cameras. We performed direct numerical simulation of a turbulent flow past a cylinder to provide training and test images, which are scalar concentration distribution fields integrated along the view direction, mimicking actual camera images. We discussed the effects of the direction in which the leaking gas flows into the camera's view and the distance between the camera and the leaking gas on the accuracy of inference. A single learner created by all images provided an inference accuracy exceeding 85%, regardless of the inflow direction or the distance between the camera and the leaking gas within the trained range. This indicated that, with sufficient training images, a high-inference accuracy can be achieved, regardless of the direction of gas leakage or the distance between the camera and the leaking gas.

**Keywords:** turbulence; passive scalar; machine learning; convolutional neural network; estimating diffusion source distance; leaking gas detection

## 1. Introduction

In petroleum and chemical plants, many measurement systems (mainly of the fixed-point type) are installed on piping and equipment to constantly monitor plant operations. Although the installation of numerous measurement systems enables closer data interval measurement, the balance between safety and economy dictates that an optimised number of instruments should be installed. In gas leak detectors, fixed-point sensors are installed according to laws and regulations. An alarm at preset gas concentrations indicates the measured values that are necessary for maintaining plant safety. In addition to the measured data, a detailed analysis of flow states based on these data enables us to understand the behavior of gas clouds in two or three dimensions, which is difficult using only fixed-point observations. This technology is expected to be useful for safe operations. In addition, as the diffusion of highly toxic substances is more dangerous when approaching a leak source, it is important to identify the leak source location based on the information from conventional gas leak detectors and analysis.

Against this background, to estimate the flow field using limited measurement data, the adjoint approach (data assimilation) has recently been studied to predict the initial turbulent flow conditions [1]. Wang et al. [2] stated that the data resolution in the streamwise or time direction should satisfy the criteria, based on the Taylor microscale in the streamwise direction. Tsukahara et al. [3] evaluated a simple method based on the Taylor diffusion theory for the turbulent transport of a passive scalar from a fixed-point source. As the Taylor diffusion theory is essentially based on the statistical properties of turbulence, their estimation from instantaneous information resulted in large errors. The time history

of the source intensity, based on sensor measurements at different locations downstream from the source by adopting an adjoint approach or data assimilation, was estimated by Cerizza et al. [4]. They showed that the estimation performance remains an issue even with multiple sensors when the scalar source is located near the wall. However, owing to the strong nonlinearity of turbulence, several issues remain for the practical application of data assimilation, including numerical stability and quick prediction.

In addition, in rapidly predicting the source of mass diffusion, sequential unsteady three-dimensional simulations based on the convective diffusion equation and the Navier–Stokes equation, together with their associated equations, are not practically applicable, given the current computer performance and computational methods.

Therefore, we focused on the application of machine learning (deep artificial neural networks) for the quick estimation of physical quantities based on observed information. Fukami et al. [5] successfully reconstructed a three-dimensional eddy flow from limited pressure data by using machine learning. Several studies have applied machine learning to predict the concentration of air pollutants in urban areas [6], detect oil spills [7], and estimate the hazardousness of leaking gases [8]. Tan et al. [9] developed a sound-source localization model, which consisted of a convolutional neural network and a regression model. Their experiments in simulated acoustic scenarios showed that the proposed model effectively estimated the angles and distances even in multiple acoustic environments under different spatial conditions. Zhou et al. [10] proposed a gas identification framework based on a sensor array for high-temperature applications. They showed the enhanced accuracy and robustness of such a framework, compared with a multilayer perceptron and support vector machine. Shi et al. [11] proposed a hybrid probabilistic deep learning model to conduct a probabilistic real-time simulation of natural gas hydrate dispersion in a deep-water marine environment. Their advanced hybrid deep learning model with variation inference and physical constraint forecast spatiotemporal concentration evolution of natural gas, compared with the point-estimation deep learning model [12].

However, to the best of the authors' knowledge, diffusion source estimation using convolutional neural networks (CNNs) has not yet been examined. Focusing on gas measurement techniques, the background-oriented schlieren method [13] and imaging methods using infrared cameras [14,15] have been developed in recent years as imaging techniques for gas leaks.

Our previous study [16] demonstrated the feasibility of applying machine learning, specifically CNNs, to estimate the diffusion distance from a point source, based on two-dimensional, instantaneous images of diffused-substance distributions downstream from the source, which was photographed by the planar laser-induced fluorescence (PLIF) method. It was found that for dye diffusion from a point source in typical parallel-plate turbulence (i.e., turbulent channel flow), the distance from the downstream image to the upstream was estimated with more than 90% accuracy. However, the flow as a test platform was limited to a single condition in terms of the Reynolds and Schmidt numbers and to a wall-bounded, fully developed turbulence. In actual engineering plants, there are various turbulent flows due to the influence of wind condition and/or obstacles, such as piping and equipment. The resulting turbulent intensity affects the degree of scalar turbulent diffusion. Thus, the applicability of our method needs to be investigated with not only a specific turbulent intensity but also under various turbulent-intensity conditions. The Schmidt number is also a key parameter for the scalar diffusion in turbulent flow. The Schmidt number of the previous experimental data [16] should be as high as O(100), which would have resulted in scalar distributions with a strong effect of turbulent diffusion and worked well for image recognition. At low Schmidt numbers, the molecular diffusion should dissipate the effective information more rapidly. In such cases, potential features in the downstream scalar distribution are lost, and the estimation of an upstream diffusion source is expected to be difficult. Thus, studies at a lower Schmidt number for typical gas are necessary to confirm the applicability of our method. In addition, the test images of our previous study were based on the concentration distribution of a plane sliced

from a certain cross-sectional area. In actual plants, the aforementioned infrared camera image is a distribution image in which the concentration is integrated along the viewpoint direction. In this image, it was assumed that small-scale concentration gradients and micro-scale fluid dynamics are not clearly captured by the integration, and relatively large-scale concentration distributions dominate the image. It is considered that the eliminated information for micro-scale fluid dynamics, which includes the scalar diffusion information, affects the inference accuracy. Therefore, the image simulating an infrared camera should be evaluated for the application.

In this study, we investigated the practical applicability of instantaneous diffusion source-location prediction using a CNN from leaking gas distribution images captured by infrared cameras. The images were obtained from direct numerical simulation of a turbulent flow with a typical Schmidt number of gas past a cylinder assuming gas leakage from surface on a piping. To consider the application in an actual plant, this study further investigated the effects of the direction in which the leaking gas flows into the camera's view and the distance between the camera and the leaking gas on the accuracy of inference. To investigate the effect of the direction of gas inflow into the camera view and the distance between the camera and the leaking gas on the accuracy of inference, we examined the effect of geometric changes (rotation, zoom-in, and zoom-out) on the generalization performance of a concentration distribution image in which the concentration is integrated along the viewpoint direction.

## 2. Methodology

To create the training images, an incompressible direct numerical simulation (DNS) was conducted using the commercial computational fluid dynamics simulation software STAR-CCM+ (ver. 2021, developed by SIEMENS, Munich and Berlin, Germany). The dimensionless governing equations are expressed as follows:

the continuity equation,

$$\frac{\partial u_i^*}{\partial x_j^*} = 0;$$

the Navier–Stokes equation,

$$\frac{\partial u_i^*}{\partial t^*} + u_j^* \frac{\partial u_i^*}{\partial x_j^*} = -\frac{\partial p^*}{\partial x_i^*} + \frac{1}{Re} \frac{\partial^2 u_i^*}{\partial x_j^* \partial x_j^*};$$

and the advection–diffusion equation,

$$\frac{\partial \phi}{\partial t^*} + u_j^* \frac{\partial \phi}{\partial x_j^*} = \frac{1}{Sc \cdot Re} \frac{\partial^2 \phi}{\partial x_j^* \partial x_j^*} + S_\phi;$$

where Re is the Reynolds number (defined later), $t$ is the time, $p$ is the pressure, and $i$ is the direction of three-dimensional Cartesian coordinate system: $x_1 = x$, $x_2 = y$, and $x_3 = z$. Einstein's summation convention is used. $S_\phi$ is the passive scalar source term and $\phi$ is the scalar value. The symbol * indicates normalisation by $u$, $\rho$, and $d$.

The computational domain is shown in Figure 1a. An image of the passive scalar behavior is shown in Figure 1b. There was a background flow inlet at $x = -80$ with Re = 1000 (made dimensionless with cylinder diameter $d$, background flow velocity $U$, and kinematic viscosity $\nu$). The cylinder was installed downstream at $80d$ from the background flow inlet ($x = 0$), and a certain amount of a passive scalar was continuously emitted from the source location, with an area of $0.01d^2$ at $y$, $z = 0$ on the cylinder surface. Training, validation, and testing images were obtained from a fully developed flow field. The same fluid and passive scalar flowed from the source at the volumetric background flow rate of $7 \times 10^{-3}$% and the Schmidt number of $Sc = 0.9$. The flow analysis meshes were approximately 9 million hexahedral meshes, and the wall meshes were set as $y^+ = 1.2$ in average (min: 0.04; max: 4.42). The Strouhal number of the Karman vortices generated in

the wake of the cylinder was confirmed to be approximately 0.2, and the computational model was thus verified. It should be noted that the spatial-discretisation accuracy of a typical DNS is of the fourth order, and STAR-CCM+ has second-order accuracy [17–19]. In this case, although its accuracy was not fully verified as the DNS standards would require, reasonable results were nevertheless obtained, thus making this simulation method suitable for preparing image data for the objective of machine learning for relatively complicated shapes, similar to this study. Figure 2a shows the mean profile of concentration at $y = 0$ in the $x$ direction at several points ($3d$, $9d$, $15d$, $21d$, $27d$, $33d$, and $39d$ from the source location), and Figure 2b denotes the root-mean-square of concentration for the same points as (a). It can be confirmed that the high-concentration passive scalar diffuses downstream, and the maximum value is near the center. The concentration becomes uniform along the x-direction, such that the concentration becomes completely uniform and no characteristic image of concentration is obtained. In this study, the concentration profile still existed at $39d$ downstream.

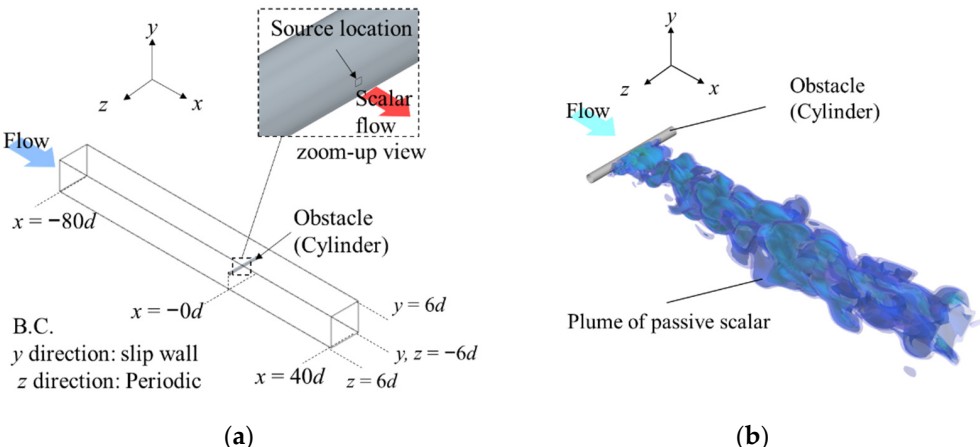

**Figure 1.** (**a**) Schematic for simulation model; (**b**) simulated plume of passive scalar.

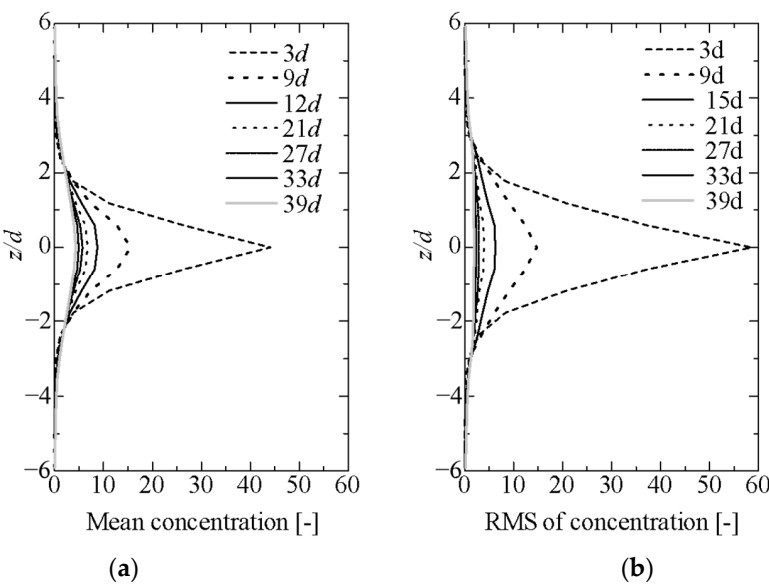

**Figure 2.** Concentration in the $x$ direction at several points ($3d$, $9d$, $15d$, $21d$, $27d$, $33d$, and $39d$ from the source location): (**a**) mean profile of concentration at the plane $y = 6d$; (**b**) root-mean-square of the concentration at the plane $y = 6d$.

The image data used in this study were concentration distributions affected by the turbulent motion of the transport medium. These image data were prepared by the

aforementioned DNS. Assuming that the infrared camera images for detecting leaking gas were realistic, the images were distribution images of $S_{integral} = \sum_{n=1}^{N_r} S_n \cdot V_n$, where $Nr$ is the number of radial divisions, $Sn$ is a scalar value, and $Vn$ is the volume. Thus, the image is a distribution image, in which the concentration is integrated along the viewpoint direction. Figure 3a shows a conceptual diagram. In this image, it can be assumed that small-scale concentration gradients and microscale features of fluid dynamics are not clearly captured by the integration, and relatively large-scale concentration distributions dominate the image. As shown in Figure 3b, images were prepared for seven points with viewing angles of 60° and 90°, and viewpoints shifted downstream from the cylinder to $0d$, $6.5d$, $13d$, $19.5d$, $26d$, $32.5d$, and $39d$. Figure 3c shows the comparison between an infrared camera sample image [20] and CFD simulation image used in this study. Infrared camera and simulation images show the small-scale concentration gradients, but micro-scale fluid dynamics are not clearly captured. Figure 4 lists sample images for each class obtained from these viewpoints.

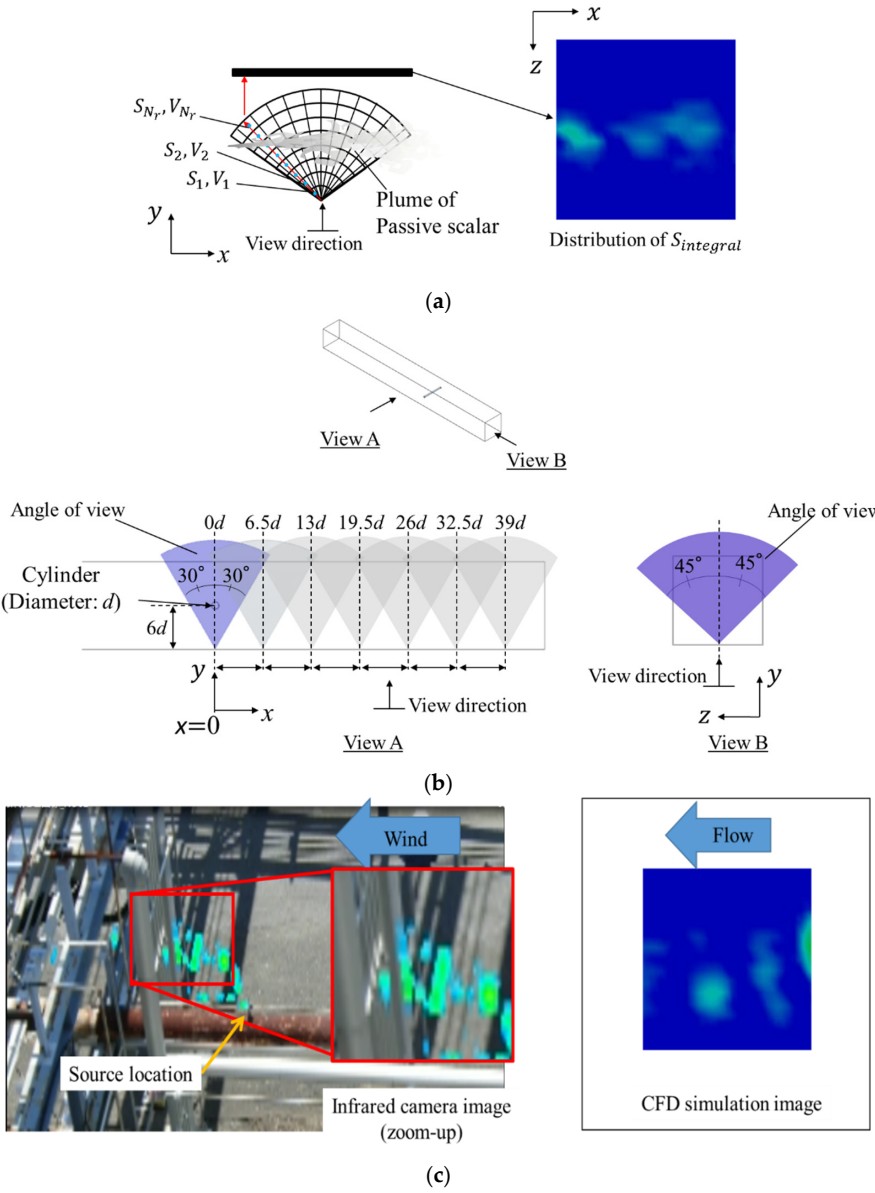

**Figure 3.** Scalar concentration images for machine learning: (**a**) conceptual diagram of distribution for $S_{integral}$: blue color indicates $S_{integral} = 0$ and red color indicates the highest passive scalar, which occurs at the source inlet; (**b**) angle of view for the image in x-y and y-z planes; (**c**) comparison of an infrared camera sample image [20] and a CFD simulation image.

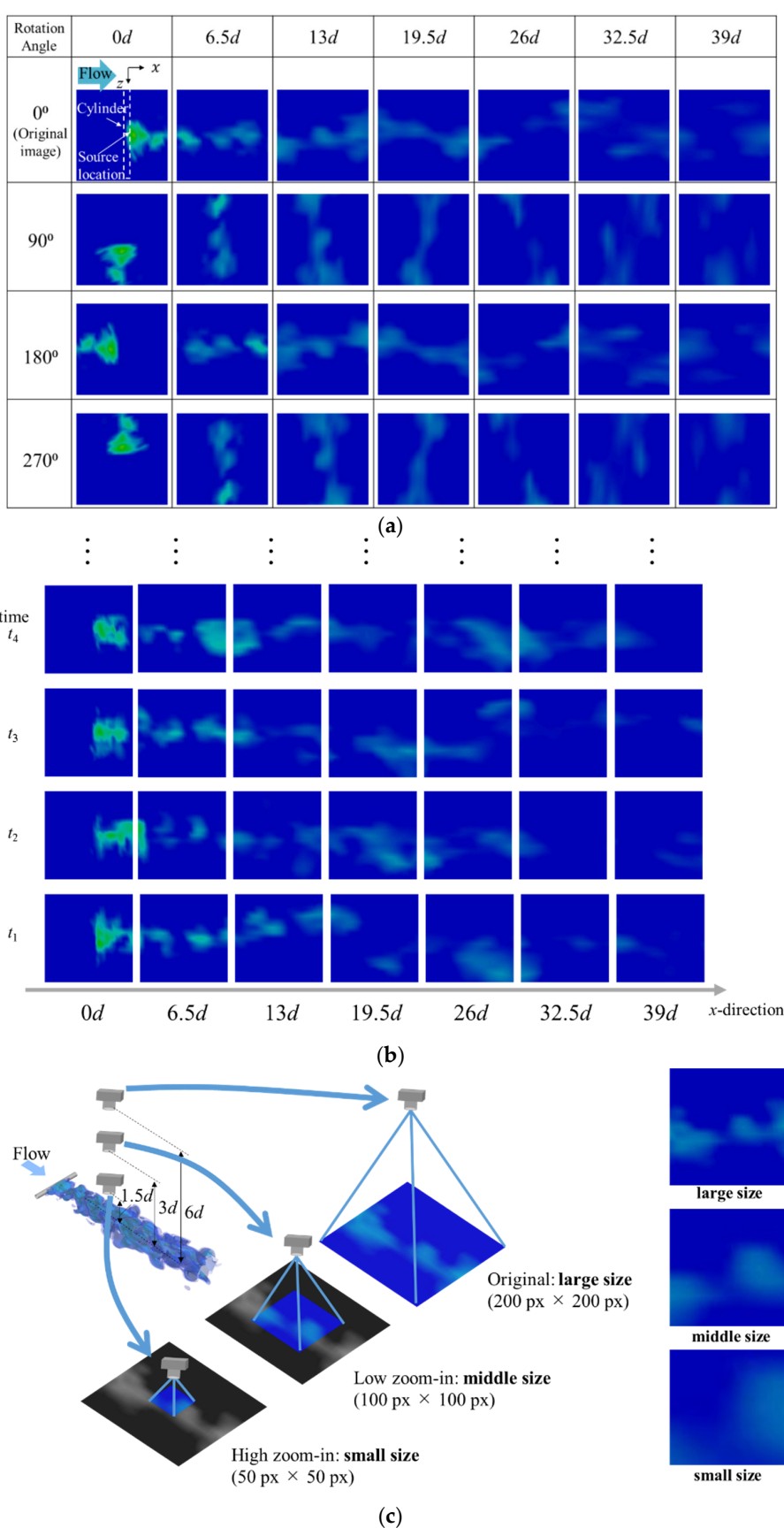

**Figure 4.** (**a**) Sample image for each class of machine learning and their rotated images; (**b**) sample images at different time instants; (**c**) conceptual diagram of each image size.

One image, as indicated in Figure 4a, has a resolution of 200 px × 200 px. Here, one pixel is equivalent to approximately $0.05d$. To check the generalization performance for rotation, the original image was at 0°, and the images were rotated clockwise to 90°, 180°, and 270°, respectively (Figure 4a). The time interval between image acquisitions was longer than the turbulence time scale, so that a variety of scalar distributions were captured in images acquired under the same conditions, as shown in Figure 4b. This provides a dataset that is less prone to overlearning. The relative positions of the camera and the gas clouds are shown in Figure 4c. To simulate the difference in the relative positions of the camera and a gas cloud, images of different sizes were created using the original image. As shown in Figure 4c, low-zoom-in (100 px × 100 px) and high-zoom-in (50 px × 50 px) images were prepared for the original size (200 px × 200 px). Hereafter, these images are called "large size," "middle size," and "small size," respectively. Table 1 lists the learners selected for this study.

**Table 1.** List of the created learners.

| | Training Image | | | | | | | | | | | |
|---|---|---|---|---|---|---|---|---|---|---|---|---|
| **Size** | **Large** | | | | **Middle** | | | | **Small** | | | |
| **Rotation** | **0°** | **90°** | **180°** | **270°** | **0°** | **90°** | **180°** | **270°** | **0°** | **90°** | **180°** | **270°** |
| Learner A | ✓ | | | | | | | | | | | |
| Learner B | ✓ | | | ✓ | | | | | | | | |
| Learner C | ✓ | ✓ | ✓ | ✓ | | | | | | | | |
| Learner D | | | | | ✓ | | | | | | | |
| Learner E | ✓ | | | | ✓ | | | | ✓ | | | |
| Learner F | ✓ | ✓ | ✓ | ✓ | ✓ | ✓ | ✓ | ✓ | ✓ | ✓ | ✓ | ✓ |

We prepared 1800 training images, 600 validation images, and 100 testing images for each class to ensure that there was no duplication. A learner was created for each image to infer an unknown testing image for the evaluation. As previous research confirmed that Inception-ResNet-v2 [21] conducted inference with a high accuracy of at least 90% [16], Inception-ResNet-v2 was used in this study to conduct a classification problem using a CNN. The architecture of Inception-ResNet-v2 is presented in Appendix A. The input sequence, activation function, and hyperparameters were set according to the values from the existing literature, and each image was resized prior to entering the network to match the input image sequence size to that of the literature. The input image was a 24-bit red–green–blue (RGB) image, and the input array was resized to be (299 × 299 × 3), where the first and second elements signified the vertical and horizontal pixels, respectively, and the third element reflected the RGB configuration. Although not shown here, preliminary research confirmed that a smaller size of the input array, compared to that of the original image (200 × 200), leads to lower inference accuracy. The current array size to enter the network allows for high inference accuracy, as reported later. Adam was used as the solver of the gradient for the mini-batch in the network.

In this study, the accuracy rate Ac was used to evaluate the inference accuracy. Ac was obtained by setting the total number of data in each class as $N_{di}$ ($i$ = 1, 2, ..., and 7) and the number of correct answers as $N_{ci}$ ($i$ = 1, 2, ..., and $N_{max}$), and the accuracy rate at each position was set as $A_{ci} = N_{ci}/N_{di}$ or the accuracy rate for all cases was set as $A_{c\_total} = \sum_{i=1}^{7} N_{ci} / \sum_{i=1}^{7} N_{di}$.

## 3. Results

To check the generalization performance for the direction in which the leaking gas flows into the camera's view, four different rotated testing images were input to each learner: a learner trained by only 0° images (Learner A), a learner trained by only 0° and 270° images (Learner B), and a learner trained by all rotated images, i.e., 0°, 90°, 180°, and 270° (Learner C). Figure 5 shows the $A_{c\_total}$ for each learner. When the same image as

the trained rotated image was used as the testing image for inference, it was confirmed that $A_{c\_total}$ was 100%. The applicability of this method to the immediate prediction of diffuse sources was confirmed by extracting the features of each location from the assumed infrared camera image.

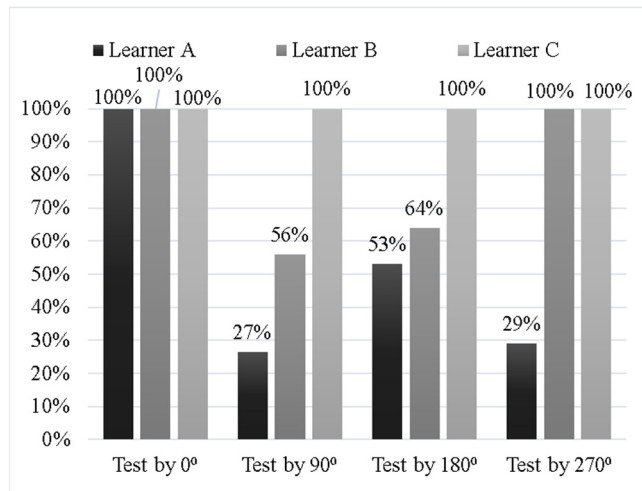

**Figure 5.** Accuracy of determination over seven classes for Learners A, B, and C by each testing image.

In contrast to Learner C, Learners A and B showed inference accuracies of 25% to 60% when the untrained rotated images were input. As CNN is robust to the parallel displacement of features by the pooling process, it was considered that the robustness to rotation was not very strong, and the reduction in inference accuracy for untrained rotated images was a reasonable result. It is known that turbulence has isotropic microscale regions, where microscale features eliminate anisotropy in fluid dynamics. It is difficult to extract rigorously the isotropic features because of the images in which the concentration is integrated along the viewpoint direction, leading to lower inference accuracy for rotational images. However, the learner that is trained with more variations in rotated images has a higher inference accuracy. For example, a learner created with images rotated by 0° and 270° (Learner B) obtained higher inference accuracy for angles (between 180° and 90°) than a learner trained only on images at 0° (Learner A). This result implies that a data augmentation method, such as rotation, successfully improves the inference accuracy for our approach without overfitting. When all rotation images (0°, 90°, 180°, and 270°) were trained (Learner C), it was confirmed that $A_{c\_total}$ was 100% correct, regardless of the rotation angle. This suggests that a deep architecture such as Inception-ResNet-v2 may result in a high inference accuracy, independent of the direction of gas cloud inflow, if training images from all angles are available.

To investigate the effect of the distance between the camera and diffused substances on inference accuracy, the dependence of the inference accuracy on the image size was determined, as shown in Figure 4c. Here, "middle size" images were used to create Learner D, and $A_c$ in each class was estimated by using the input image sizes "large size" and "small size", which were different from the training images. The confusion matrices are shown in Figure 6a–c. For the middle-sized image, which had the same size as the training image, the accuracy was higher than 95% at all locations (Figure 6b), while for the large-sized image, the diffusion source distance tended to be underestimated upstream from the correct solution (Figure 6a). For small-sized images, the diffusion source distance was overestimated downstream from the correct solution (Figure 6c). This is consistent with the fact that the smaller the size of the image, the greater the diffusion of the substance in the longitudinal direction of the cylinder (z-direction) along the downstream direction; thus, the greater the diffusion of the substance, the smaller the size of the image. Conversely, in larger images, it was recognised that the substance was not diffused. From the results, the

*z*-direction diffusion of the substances against the training image size was also inferred as a feature.

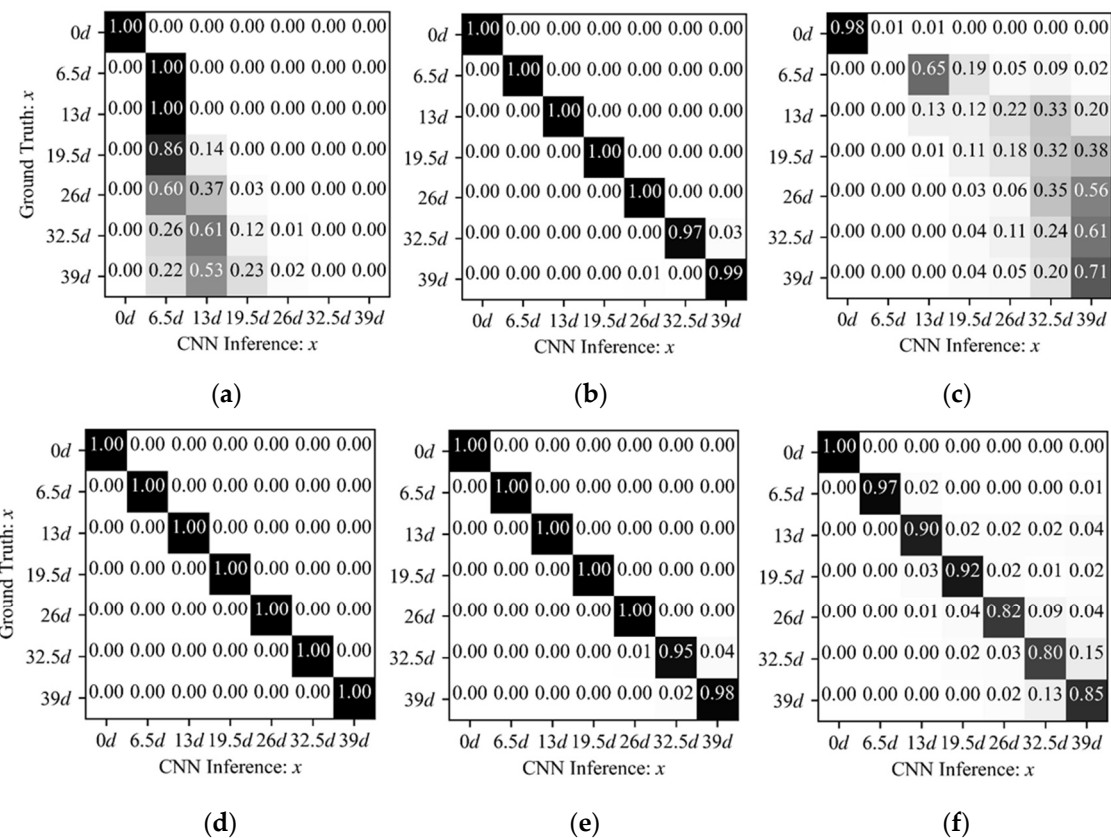

**Figure 6.** Accuracy of determination for image angle $0°$: (**a**) Learner D tested by "large size" images; (**b**) same as (**a**), but tested by middle size; (**c**) same as (**a**), but tested by small size; (**d**) Learner E tested by "large size" images; (**e**) same as (**d**), but tested by middle size; (**f**) same as (**d**), but tested by small size.

A single trainee (Learner E) was created with all three sizes as training images, and the confusion matrix for each size is shown in Figure 6d–f. The inference accuracy was approximately 80% for small images at 26*d* downstream, and more than 90% for the other locations. This suggests that, similar to image rotation, if sufficient training images can be prepared for the image size, a high inference accuracy can be obtained for the application camera, regardless of the distance between the gas cloud and the camera. However, the slight drop in inference accuracy downstream from 26*d* for the small size may have occurred because the camera was significantly close (the gas cloud was magnified) and did not adequately capture the scale that is characteristic of turbulent mixing [10]. To confirm the characteristic scale in the $S_{integral}$ distribution, the autocorrelation coefficient, $R_{BB}$, for the *x*-direction of luminance relative to the image's center, is shown in Figure 7.

$$R_{BB}(\mathbf{r}) = \frac{\overline{B'(\mathbf{x})B'(\mathbf{x}+\mathbf{r})}}{\overline{B'(\mathbf{x})B'(\mathbf{x})}},$$

where $\mathbf{r}$ is a spatial two-point distance vector and $B(\mathbf{x})$ is the brightness fluctuation value of each pixel calculated from the image data of the distribution of $S_{integral}$. The overbar denotes the ensemble average. The fluctuating component $B'(\mathbf{x})$ denotes the brightness value of each pixel minus the average brightness, which was obtained in advance from all the image data points. As the autocorrelation coefficient is based on the image's center point, half of the image width pixels are at the edge of the image. Here, $R_{BB} \approx 0.2$ at 100 px

downstream for the large size and $R_{BB} \approx 0.4$ at 50 px downstream for the middle size. However, for small sizes downstream from more than $13d$, $R_{BB} \approx 0.7$ at the edge of the image (25 px), which is relatively higher. This means that a small-sized image captured only a part of the fluid motion dynamics, but not all motions and, therefore, it is difficult to make inferences from such limited information. Therefore, a large-sized image that can capture the overall gas cloud is necessary to improve the accuracy of inference.

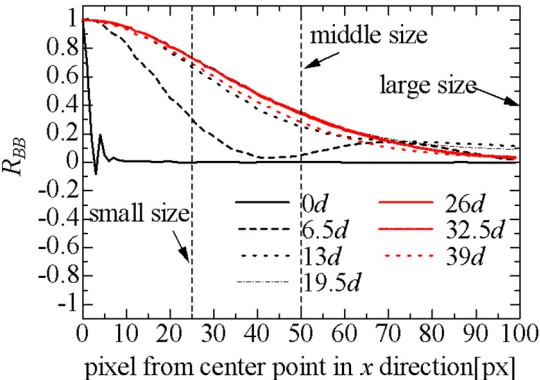

**Figure 7.** Autocorrelation coefficient of the brightness fluctuation value of each pixel, calculated from the image data of distribution for $S_{integral}$.

To improve the generalizability in both rotation direction and distance between the camera and leaking gas, a single trainee (Learner F) was created by training a total of 12 different images, where each image was a combination of four different rotation images ($0°$, $90°$, $180°$, and $270°$), and three different image sizes: small, middle, and large. The $A_{c\_total}$ inferred for each of the 12 unknown images is shown in Figure 8. The accuracy rate exceeded 85%, regardless of image rotation and size. This indicated that when sufficient training images are prepared by data augmentation, the inference accuracy is high, regardless of the direction of the leaking gas flow into the camera's field of view or the distance from the camera.

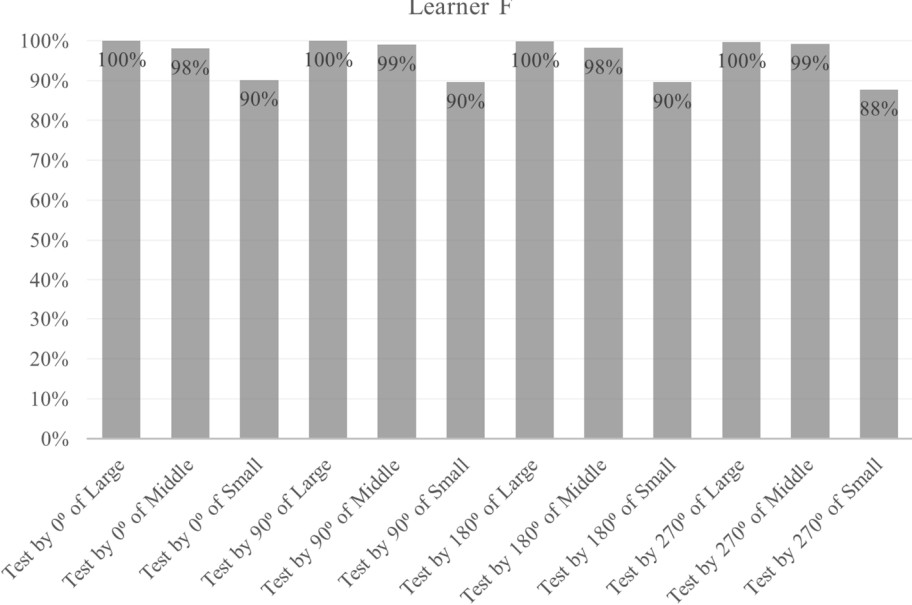

**Figure 8.** Accuracy of determination over seven classes for Learner F.

## 4. Conclusions

We investigated the applicability of CNNs for predicting the diffusion sources of turbulent substances using leaking gas detection images from infrared cameras. The image

data prepared by DNS are concentration distributions affected by the turbulent motion of the transport medium. A concentration distribution image was used, in which the concentration was integrated along the view direction, assuming an actual camera image. In this image, the small-scale concentration gradients and small-scale features of fluid dynamics are difficult to capture clearly because of the integration along the view direction, and relatively large-scale concentration distributions dominate the image. The estimation of the distance from the leakage source was performed as a classification problem, divided into seven classes according to the distance downstream from the leakage source.

The effects of the direction in which the leaking gas flows into the camera's view and the distance between the camera and the leaking gas on the accuracy of the inference were examined. The images were prepared for data augmentation by rotating and scaling the original images. The inference accuracy for unknown images was examined.

For the rotated images, 100% accuracy was obtained for the same rotated image as the training image. However, for rotated images that were different from the training image, the inference accuracy was 25–60%, thereby resulting in poor generalization performance.

As CNN is robust to the parallel displacement of features by the pooling process, it was considered that the robustness to rotation was not very strong, and the reduction in inference accuracy for untrained rotated images was a reasonable result. It is known that turbulence has isotropic microscale regions, where microscale features eliminate anisotropy in fluid dynamics. It is difficult to extract rigorously the isotropic micro-scale features because of the images in which the concentration is integrated along the viewpoint direction, leading to lower inference accuracy for rotational images. However, when all rotation images ($0°$, $90°$, $180°$, and $270°$) were trained, it was confirmed that $A_{c\_total}$ was 100% correct regardless of the rotation angle.

To investigate the effect of the distance between the camera and diffused substances on the inference accuracy, the inference accuracy for different image sizes was examined. For images that were different in size from the training image, the inference accuracy was lower, resulting in a poor generalization performance, similar to the image rotation case. However, it was found that a high inference accuracy could be obtained if the data were trained with all images, regardless of the distance between the gas cloud and the camera.

To improve the generalizability in both rotation direction and distance between the camera and leaking gas, a single trainer was created by training all images, and the inference accuracy exceeded 85%, regardless of the image rotation and size. This indicated that when sufficient training images are prepared by data augmentation, the inference accuracy is high, regardless of the direction of the leaking gas flow into the camera's field of view or the distance from the camera.

In the future, when the so-called digital twin is realized and training data can be obtained from the digital simulation data of the plant, many leakage scenarios can be run in such simulations, and a trainer can be created based on snapshot images obtained from camera arrangements in an actual plant. This study showed that a data augmentation method, such as rotation and image size, successfully improves the inference accuracy for our approach without overfitting. This implies that by utilizing data augmentation for image data, it may be possible to improve inference accuracy not only for a specific plant situation. For further practical applications for example, disaster prevention, pollution control, etc., the gas diffusivity caused by actual fluctuations in wind conditions should be considered.

**Author Contributions:** Conceptualization, T.I. and T.T.; methodology, T.I.; analysis, T.I.; investigation, T.I. and T.T.; writing—original draft preparation, T.I.; writing—review and editing, M.I. and T.T; visualization, T.I. All authors have read and agreed to the published version of the manuscript.

**Funding:** This work was partially supported by the Japan Society for the Promotion of Science (JSPS), Grant-in-Aid Scientific Research (S) and (A): Grant Number 21H05007 and 18H03758.

**Institutional Review Board Statement:** Not applicable.

**Informed Consent Statement:** Not applicable.

**Data Availability Statement:** Not applicable.

**Conflicts of Interest:** The authors declare no conflict of interest.

## Appendix A

Architecture for Inception-ResNet-v2 is shown in Figure A1a. Inception-ResNet-v2 implements 164 layers by adopting residual inception blocks (Figure A1b), which was developed in ResNet (2015) [22] as the breakthrough. Three types of residual inception blocks (A, B, and C) were introduced, and multiple layers were created by repeating A, B, and C five, ten, and five times, respectively. With the residual inception block, efficient learning was possible, even in deep networks, via the dimensionality reduction given by inserting a $1 \times 1$ convolution layer, convolutions of different sizes in the branched network, and the operation of passing input directly to the next layer by shortcutting the bias.

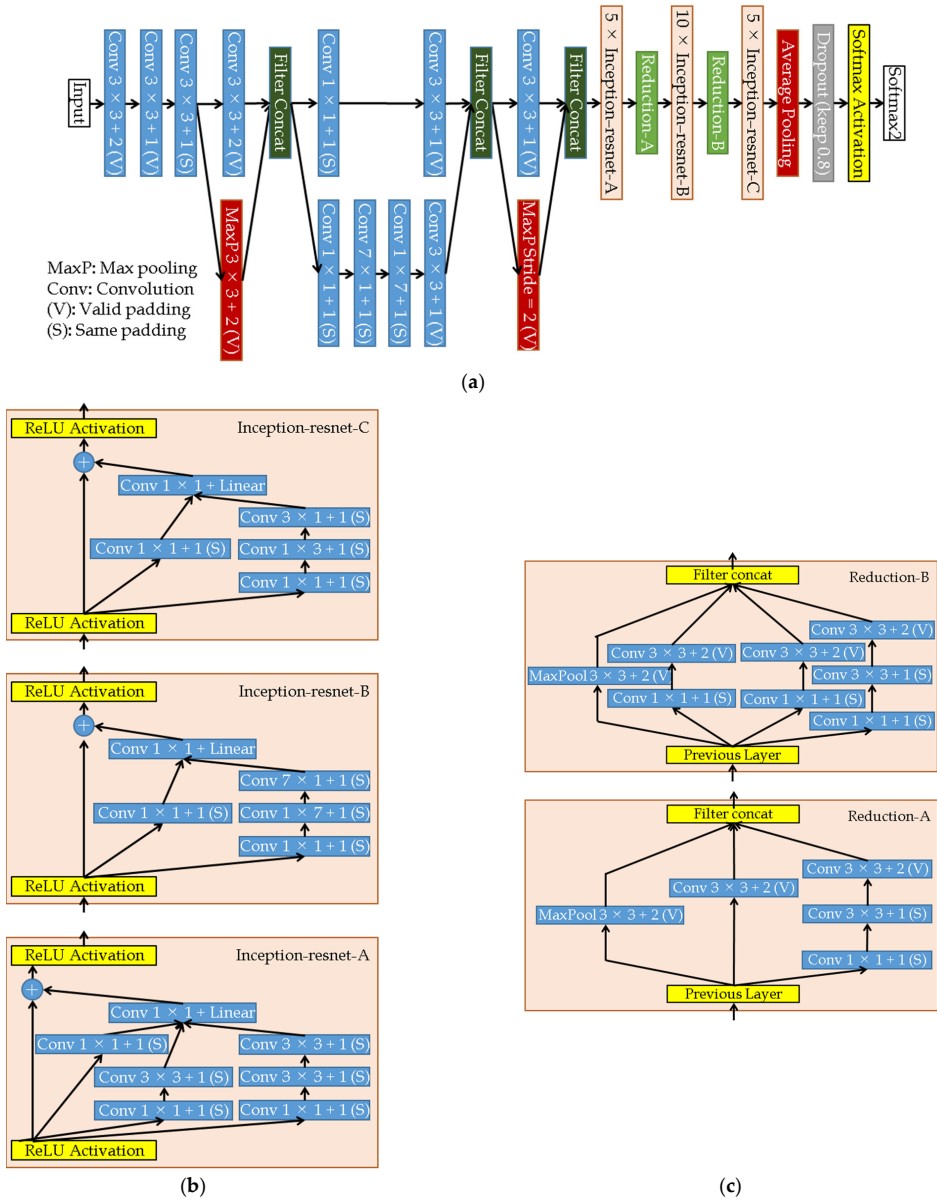

**Figure A1.** Schematic of Inception-ResNet-v2 architecture [22] referred to in our previous paper [16]: (**a**) overall network; (**b**) Inception-Resnet blocks: "-A" is repeated five times, "-B" is ten times, and "C" is five times; (**c**) Reduction-A and Reduction-B.

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
