# Peer review of "Applicability of Convolutional Neural Network for Estimation of Turbulent Diffusion Distance from Source Point"

_processes, doi:10.3390/pr10122545_

Round 1

Reviewer 1 Report

This is an excellent manuscript. I recommend it.

Reviewer 2 Report

This manuscript investigates the application of convolutional neural network to estimate the turbulent diffusion distance from source point, which change the task to a classification task and achieve good inference accuracy. However, it is very similar to the author's previous work [12], both of which use CNN to predict the distance from source point. The author should show more details in differences between the two works except the input images.

1.      In Section Introduction, the motivation should be more explicit. It is easy to get confused about the essential difference between this manuscript and the previous study mentioned by the author.

2.      In Section Methodology, the training images are all created in the same flow field and only one example image of each class is shown in Figure 4. Are the images in the training set very similar to each other? It is necessary for this manuscript to show several images of each class to prove the reliability and validity of the dataset.

3.      In Section Results, as a common data augmentation method in computer vision, rotation is expected to improve the inference accuracy. The author forgot to analyze this possibility in the discussion of Figure 5.

4.      In Figure 8, I think the manuscript should indicate the specific value of each result.

5.      The last paragraphs of Section Results and Section Conclusions are almost repetitive. The author should rewrite one of them or present something more meaningful.

6.      I think the number of references in this manuscript is not enough, the author should introduce more works and supply more references.

Reviewer 3 Report

This paper explores the applicability of CNN for estimating turbulent diffusion distance from source point under different camera’s view direction and taking distance. This work is helpful for integrating AI based model and infrared imaging for gas leakage detection. However, this paper further need to be revised before accepting.

Other comments:

Q1) The keywords further need to be improved. Suggest deleting “inverse problem”, “channel flow” and “source prediction”.

Q2) Please add references regarding estimating the flow field using limited measurement data based on machine learning approaches. Examples are as follows:

1. Real-time natural gas release forecasting by using physics-guided deep learning probability model. Journal of Cleaner Production, 368, 133201.

2.  Probabilistic real-time deep-water natural gas hydrate dispersion modeling by using a novel hybrid deep learning approach. Energy, 219, 119572.

Q3) The figure 1 demonstrates that this range of z-axis should be -6d to 6d, which is inconsistent with the figure 2. what does the value of z-axis represent in the figure 2?

Q4) Page 4 Line 163: do the resizing operation for images affect the concentration distribution and so affect the classifier’s model performance.

Q5) Please explain the similarity between simulation images by STAR-CCM+ and the real images by infrared cameras.
